# Finetuning from Offline Reinforcement Learning: Challenges, Trade-offs and Practical Solutions

## Abstract

Offline reinforcement learning (RL) allows for the training of competent agents from offline datasets without any interaction with the environment. Online finetuning of such offline models can further improve performance. But how should we ideally finetune agents obtained from offline RL training? While offline RL algorithms can in principle be used for finetuning, in practice, their online performance improves slowly. In contrast, we show that it is possible to use standard online off-policy algorithms for faster improvement. However, we find this approach may suffer from *policy collapse*, where the policy undergoes severe performance deterioration during initial online learning. We investigate the issue of policy collapse and how it relates to data diversity, algorithm choices and online replay distribution. Based on these insights, we propose a conservative policy optimization procedure that can achieve stable and sample-efficient online learning from offline pretraining.

## 1 Introduction

Offline reinforcement learning (ORL) (Lange et al., 2012; Levine et al., 2020) considers the problem of learning policies from fixed datasets without requiring additional interaction with the real environment. ORL has the potential to enable sample-efficient learning in applications such as healthcare and robotics as it does not require potentially expensive interaction with the real environment. However, recent work (Foster et al., 2021) suggests it may be challenging for pure ORL approaches to learn optimal behavior using only offline data, e.g., if the relevant parts of the state-action space are not well represented in the offline dataset. When the offline policies are not optimal for the real environment, *finetuning* with additional online data can enable the agents to achieve stronger performance (Nair et al., 2021; Lee et al., 2021). For finetuning to be practical, however, it is important to ensure that the online finetuning procedure improves fast from online data. Unlike in supervised learning, where there are more established approaches to pretraining and finetuning, how to best finetune RL agents trained from prior experience remains less well understood.

In this paper, we study how to improve offline pre-trained policies with a small amount of additional online interactions. One may hope to deploy the offline policy to collect more data and reuse the same algorithm for offline learning and finetuning. However, existing studies and our findings indicate that finetuning with RL algorithms designed for offline learning converges slowly with additional online data (Nair et al., 2021). An alternative approach to re-using the offline RL algorithm for online finetuning is to use a different online off-policy RL algorithm for online finetuning. Since recent online off-policy algorithms (Fujimoto et al., 2018; Lillicrap et al., 2016; Abdolmaleki et al., 2018; Haarnoja et al., 2018) have demonstrated strong performance and good sample efficiency, we should expect that additional pretraining with offline RL would allow this approach to enable sample-efficient finetuning. We show that using a standard off-policy RL algorithm can work well for online finetuning. However, we also observe that sometimes finetuning with online off-policy algorithms can lead to *policy collapse*, where the policy performance degrades severely during initial online training. These observations motivate us to investigate the challenges in different strategies for finetuning from offline RL. Towards this goal, we analyze the trade-offs in choices of algorithms and whether/how to use offline data for online finetuning. We present several approaches to finetuning and discuss their merits and limitations based on empirical observations on standard offline and online RL benchmarks. Based on our observations, we conclude that effective online finetuning from offline RL may be achieved with more robust online policy optimization

and present a constrained policy optimization extension to the TD3 algorithm, which we call conservative TD3 (TD3-C), that empirically helps stabilize online finetuning, thereby addressing the issue of policy collapse.

## 2 Background

**Off-policy online reinforcement learning** We consider reinforcement learning (RL) in a Markov Decision Process (MDP) defined by the tuple $(\mathcal{S}, \mathcal{A}, p, p_0, r, \gamma)$. Off-policy RL algorithms learn a policy $\pi_\theta$ with experience generated by a different policy $\mu$. Example algorithms include Deep Deterministic Policy Gradient (DDPG) (Lillicrap et al., 2016) and Twin Delayed Deep Deterministic Policy Gradient (TD3) (Fujimoto et al., 2018). These deep off-policy algorithms learn an approximate state-action value function $Q_\phi$ and deterministic policy $\pi_\theta$ by alternating policy evaluation and improvement. During policy evaluation, we learn an approximate state-action value function $Q_\phi$ by minimizing the Bellman error

$$\phi^* = \arg\min_\phi \mathbb{E}_{s,a,r,s' \sim \mathcal{B}} \left[ (Q_\phi(s,a) - (r + \gamma Q_{\phi'}(s', \pi_{\theta'}(s')))^2] \right], \tag{1}$$

where $Q_{\phi'}$ and $\pi_{\theta'}$ are the target critic and policy networks used to stabilize TD learning with function approximation. During policy improvement, the policy parameters $\theta$ are updated to maximize the current state-action value function

$$\theta^* = \arg\max_\theta \mathbb{E}_{s \sim \mathcal{B}} \left[ Q_\phi(s, \pi_\theta(s)) \right]. \tag{2}$$

**Leveraging demonstrations in RL** In addition to the standard reinforcement learning problem definition, we consider the setting where we have access to an additional behavior dataset. for example, a dataset consisting of environment transitions $\mathcal{B} = \{(s_i, a_i, r_i, s_{i+1})\}_{i=1}^N$ of experience collected by an unknown behavior policy $\mu$ in the same MDP.

The idea of leveraging demonstrations in reinforcement learning is not new. Behavior cloning (Pomerleau, 1989) is a supervised learning method that trains the policy to mimic the behavior policy induced by the demonstration dataset consisting of states and corresponding actions. Imitation learning (Schaal, 1999) aims at learning policies that produce trajectories close to those in the demonstration dataset; however, actions are typically not observed in imitation learning. In addition, expert demonstrations have been shown to be effective in accelerating learning in sparse reward problems (Vecerik et al., 2018). These approaches require the demonstrations to consist of expert or high-quality trajectories.

**Offline RL with online finetuning** More recently, offline RL has received increasing attention due to its potential for scaling RL to large-scale, real-world applications. Unlike imitation learning or behavior cloning, offline RL utilizes reward-annotated datasets and can in principle learn an optimal policy given sub-optimal logged trajectories by "stitching" together good behaviors.

In principle, we can apply off-policy algorithms, such as DDPG or TD3, to learn from a fixed dataset; however, in practice, they are ineffective when learning from fixed offline datasets due to extrapolation errors (Fujimoto et al., 2019). This manifests as learning massively over-estimates state-action values during offline training. Recently, many deep offline RL algorithms Kostrikov et al. (2022); Kumar et al. (2020); Wang et al. (2020); Fujimoto et al. (2019) have been proposed to reduce the extrapolation error. These algorithms modify standard deep off-policy actor-critic algorithms and constrain policy learning to be supported by the dataset, which helps reduce extrapolation error. Approaches to reduce extrapolation error include, but are not limited to, value-constrained methods that learn conservative value estimates for out-of-distribution actions (Kumar et al., 2020) and policy-constrained methods that regularize the policy to not deviate significantly from the empirical behavior policy (Nair et al., 2021; Kostrikov et al., 2022; Wang et al., 2020). These offline algorithms can learn competitive policies, surpassing the performance of the empirical behavior policies in the dataset.

Typical offline RL algorithms focus on learning from the fixed dataset and do not permit access to an online environment; their sole focus is to learn a good policy using the offline dataset only (Levine et al., 2020). In this paper, we are interested in using additional online environment interactions to improve the agent's

performance further. Concretely, we consider the setting where we first pretrain a policy using offline RL and then further finetune it online, similar to the scenario considered in (Kostrikov et al., 2022; Lee et al., 2021; Nair et al., 2021). Although offline RL algorithms can be used for finetuning with additional online data, previous work (Nair et al., 2021; Lee et al., 2021) found that many offline RL algorithms improve slowly when given additional data. For example, (Lee et al., 2021) found that online finetuning with the offline RL algorithm CQL results in a slight improvement with the addition of online data. In this paper, we are interested in understanding the sub-optimality in performance and finding better strategies for finetuning agents obtained from offline pretraining.

## 3 Challenges in Finetuning from Offline RL

In this section, we empirically study and analyze the challenges in performing online finetuning after pretraining with offline RL. Our analysis builds on top of MuJoCo (Todorov et al., 2012) tasks in the D4RL benchmark suite (Fu et al., 2021). We consider datasets from the walker2d, halfcheetah, hopper, and ant tasks. For each task, we perform finetuning given the corresponding medium and medium-replay datasets. The medium datasets consist of transitions collected by the evaluation policy of an early-stopped, sub-optimal agent. Medium-replay datasets refer to the transitions stored in the replay buffer of an early-stopped agent. Both types of datasets contain transitions that enable the agent to acquire medium performance. The medium quality dataset includes only transitions from the learned policy, which, on average, have higher quality than the medium-replay datasets but lack diversity.

### 3.1 Experimental Setup

For offline pretraining, we use TD3-BC (Fujimoto and Gu, 2021). TD3-BC is a policy-constrained offline RL algorithm that extends the TD3 algorithm by including an additional behavior cloning (BC) regularizer in the policy improvement step to encourage the policy to stay close to the behaviors in the offline dataset. Specifically, TD3-BC augments the policy optimization objective in Equation (2) with an additional penalty $(a - \pi_\theta(s))^2$, such that

$$\theta^* = \arg \max_\theta \mathbb{E}_{s,a \sim \mathcal{B}} \left[ \lambda Q_\phi(s, \pi_\theta(s)) + (a - \pi_\theta(s))^2 \right], \tag{3}$$

where $a$ is the action stored in the offline dataset, and $\lambda \geq 0$ modulates the relative strength of the original policy optimization objective and the behavior cloning regularization. We selected TD3-BC because of its good empirical performance on the MuJoCo locomotion benchmarks. Further, it is also easy to compare TD3-BC with its off-policy counterparts, TD3, for finetuning performance since switching from TD3-BC to TD3 requires only removing the BC penalty, keeping all other hyper-parameters fixed.

During online finetuning, we load the weights for the neural networks obtained from offline training and use either TD3 (Fujimoto et al., 2018) or TD3-BC as the finetuning algorithm. Hyperparameters are chosen to be the same as the offline setting. In both cases, we pretrain the actor and the critic offline for $500K$ iterations and then train online for $200K$ environment steps. We perform one gradient step after every transition in the online environment during online finetuning. Therefore, one learner step after the pretraining stage corresponds to one step of online environment interaction. Note that the online training setup is different from the online batch setting considered, for example in (Lee et al., 2021; Nair et al., 2021), but is closer to the standard online benchmark setting used in (Fujimoto et al., 2018).

Our experiments are implemented in JAX (Bradbury et al., 2018) based on DeepMind's Acme RL library (Hoffman et al., 2020) and JAX ecosystem (Babuschkin et al., 2020). We use an internal computing cluster for the experiments; each job has access to a single GPU. Reproducing the experimental results in this paper takes approximately 600 GPU hours. Code for reproducing our experiments will be open-sourced.

### 3.2 The Effect of Online Algorithms During Finetuning

We start by analyzing how the choice of online algorithms impacts finetuning performance. For this experiment, we use either TD3 or TD3-BC as the online algorithms for finetuning. Following the finetuning

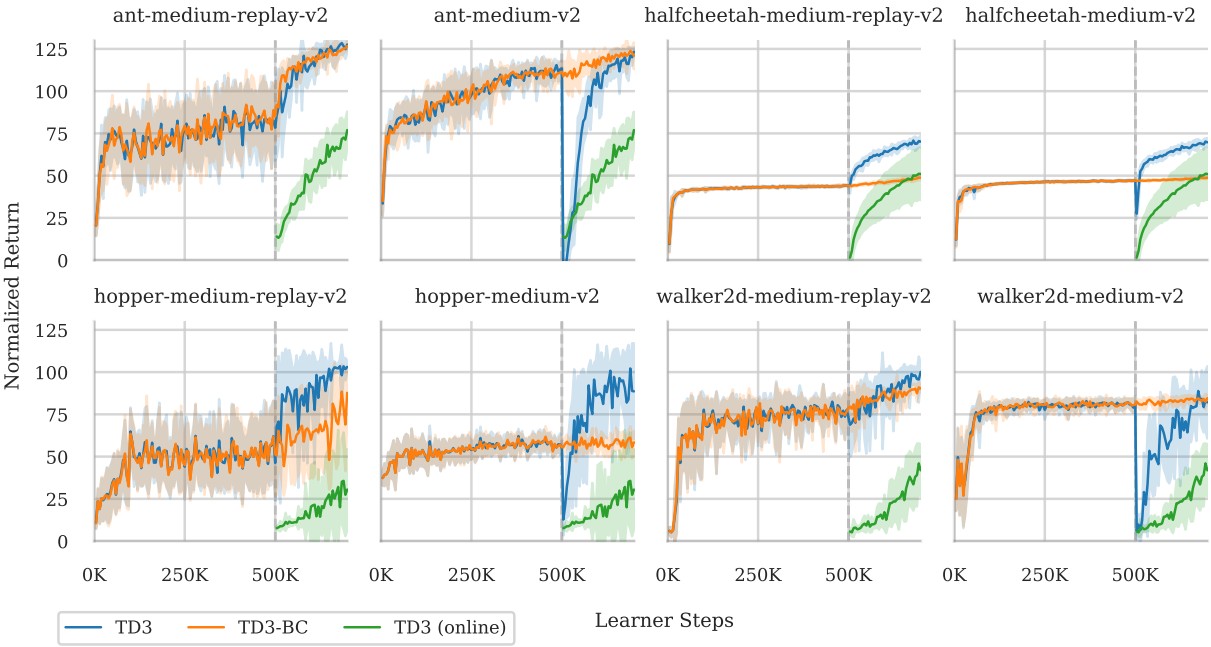

Figure 1: Comparison of TD3, TD3-BC for online finetuning on the D4RL benchmark suite. The agents are pretrained for 500K steps with TD3-BC before finetuning with additional online data for 200K environment steps. We perform one gradient step for every environment step, so the number of learner steps after 500K corresponds to the number of environment steps taken during finetuning. TD3-BC (orange) improves slowly with finetuning compared to TD3. However, TD3 (blue) shows policy collapse during initial finetuning. Policy collapse is more observable in the medium datasets than in the medium-replay datasets, which are more diverse. We also include results for training a TD3 agent (green) without any offline pretraining and it performs worse than agents with offline pretraining, demonstrating that offline pretraining is useful in accelerating sample efficiency in online learning. Results are averaged with ten random seeds.

protocol in (Zheng et al., 2022; Kostrikov et al., 2022; Nair et al., 2021), we also incorporate the offline data during finetuning by initializing the online replay buffer with transitions from the offline dataset. The transitions are sampled uniformly during both offline pretraining and online finetuning. Figure 1 shows the comparison between the different choices of online algorithms on evaluation performance. The results reveal a few interesting findings.

**Offline RL algorithms improve more slowly compared to their online counterparts.** Finetuning online with TD3-BC improves slowly compared to using TD3. This suggests that ORL algorithms, which constrain the target policy to be close to behavior policy, may improve more slowly than their standard off-policy counterparts. While we restrict our comparison to using the TD3 algorithm as the base RL algorithm, previous work (Lee et al., 2021; Nair et al., 2021) shows similar findings for other ORL algorithms.

**Online finetuning with off-policy algorithms suffers from policy collapse.** While online finetuning with TD3 achieves a better evaluation score compared to TD3-BC, there is a noticeable training instability for some datasets in the early stages of online finetuning. This is illustrated by the sudden drop in performance as finetuning starts (i.e., 500K learner steps). This is distinct from the fluctuations in performance that is typical in deep off-policy RL algorithms. This phenomenon is sometimes referred to as *policy collapse.* Policy collapse happens as the critic is inaccurate when finetuning starts and is over-optimistic on novel states encountered early during finetuning. Nevertheless, finetuning with TD3 after 200K environment steps always rivals or surpasses TD3-BC. Note that policy collapse is not a specific problem for TD3: (Nair et al., 2021;

Lee et al., 2021) consider offline pretraining with CQL followed by finetuning with SAC, and the training instability can also be observed in their experiments.

**Policy collapse is more severe when the diversity of the dataset is lower.** The extent of policy collapse varies across domains and dataset qualities and is more noticeable as the diversity of the dataset decreases. Although the offline performance on the medium and medium-replay datasets are comparable, finetuning from agents pretrained with TD3 on the medium datasets is more unstable. This is expected since the medium datasets are less diverse than the medium-replay datasets. Notice that the rate at which TD3 recovers its original performance also varies across the datasets, and it is more difficult to recover the performance when pretrained on less diverse datasets.

### 3.3 The Effect of Offline Data During Finetuning

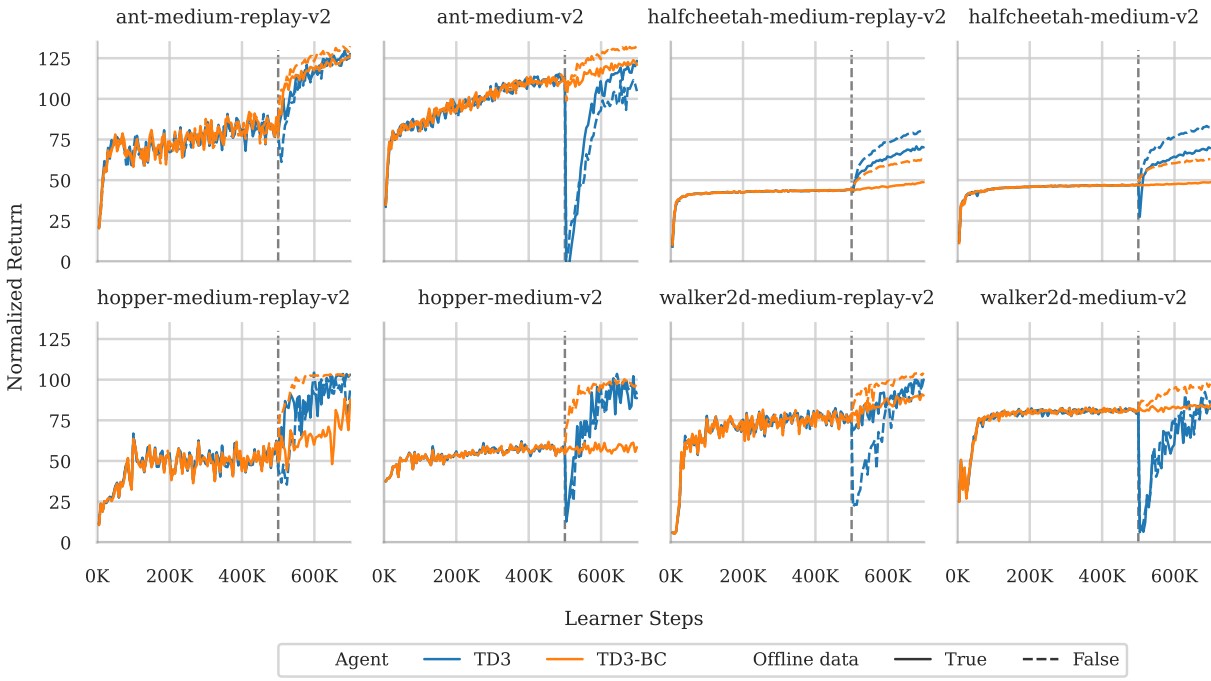

Figure 2: Effect of using offline datasets for online finetuning. We compare what happens if we do not initialize the online replay with transitions from the offline dataset. TD3-BC enjoys significant improvement if we do not sample offline transitions during online finetuning. TD3 exhibits policy collapse independent of whether offline data is utilized during finetuning. The standard deviation is ignored in the plots for better visibility.

In Section 3.2, we followed the finetuning protocol used in (Kostrikov et al., 2022; Zheng et al., 2022), which loads the offline data into the online replay before finetuning starts. In the following, we consider what happens if we do not utilize any offline data during finetuning. In this case, we are just initializing the online policy and critic networks with the pretrained weights. Figure 2 shows the effect of loading the offline data into the online replay before finetuning begins. TD3-BC remains stable during finetuning even when offline data is not utilized. Discarding the offline data allows it to obtain higher sample efficiency and rival or even outperform online finetuning on six out of the eight datasets. This indicates that constraining the policy outputs to be close to actions stored in the replay offers improved stability even when the replay data is collected purely from the online environment. At the same time, however, this regularization via BC may hurt finetuning if the goal is to maximize performance improvement given a small number of online interactions. TD3-BC can still perform worse than online RL algorithms, as evident from the lower sample efficiency in the halfcheetah datasets. TD3 can also obtain higher performance without utilizing offline data, and finetuning

is more stable with pretraining on medium-replay datasets than medium datasets. The only exception is walker2d-medium-replay, where not initializing with offline data results in policy collapse. However, for the less diverse medium datasets, policy collapse happens independent of whether offline data is reused, and the drop in performance happens immediately as finetuning starts.

### 3.4 Summary of Empirical Observations

In the following, we summarize our observations of the empirical study.

- The degree of the conservativeness of the RL algorithms has a substantial impact on the data efficiency during finetuning, where the goal is to maximize the improvement compared to the pretrained policies. Mitigation strategies that reduce extrapolation errors during offline training can also benefit online finetuning by improving the stability of the online training. However, this may come as a trade-off for online finetuning sample efficiency. Using online RL algorithms may result in a faster rate of improvement but can be less stable, and such an instability may be undesirable if the online sampling budget is limited. The degree of instability depends on the properties of the underlying MDP and the diversity of the offline dataset. When the offline dataset has enough diversity, the errors encountered during online finetuning will be mild and may not be significant enough to collapse a good pretrained policy. However, if the value estimate is inaccurate during finetuning, then policy optimization algorithms that maximize the erroneous critic estimate will result in policy collapse.

- The online replay sampling distribution plays an important role. The two approaches we considered, namely whether we initialize the online replay with offline datasets, present different trade-offs. However, when the online replay is initialized with the offline dataset, the sampled transitions during the initial finetuning period should resemble those seen during offline training. For policy-constrained offline RL algorithms such as TD3-BC, the additional policy constraint will regularize the online training policy heavily towards sub-optimal behaviors in the offline dataset. This explains why TD3-BC improves faster when we completely discard offline data during finetuning. However, online algorithms that do not incorporate any additional constraints suffer from training instability.

- Initializing the online replay with offline data followed by uniform sampling during finetuning may present additional issues that influence the stability of online algorithms such as TD3. When the offline dataset is large, during the initial period of finetuning, samples from the replay buffer would consist mainly of transitions from the offline dataset since the amount of new samples is relatively small compared to the number of offline samples. This can create a similar effect as learning offline during online finetuning. Since TD3 is not designed for offline learning, it may overestimate the values during initial finetuning. We discuss this issue in more detail in Section 4. A solution that prevents the offline samples from crowding out the online replay is to fix the ratio of offline and online samples during finetuning. This has the advantage of ensuring that the agent immediately uses online interactions collected by the agent during finetuning, decoupling the rate at which online transitions are sampled from the size of the offline dataset. The results are shown in Appendix C.1. In this case, TD3 still suffers from policy collapse and can only be explained by an inaccurate critic.

- When the offline data is not used to initialize the online replay, all transitions sampled during finetuning will consist purely of online interactions. The critic may encounter more transitions not seen during pretraining, making stable finetuning difficult. The agent may also suffer from the risk of catastrophic forgetting since the agent would not be able to recall any bad behavior it has experienced during offline learning. While we have presented experimental results for the two extreme settings, it is unlikely that either would be the best solution for practical applications. While (Lee et al., 2021) explored using prioritized replay that learns to mix the offline and online samples that allow utilization of offline samples during online learning, the issue of policy collapse can still happen with their approach. Therefore, our experiments suggest that if the online algorithm does not explicitly address issues that prevent extrapolation errors in the critic from crippling the policy, it is unlikely that a different choice of the replay sampling strategy can help prevent policy collapse.

Our observations suggest that regularizing the policy optimization during finetuning is useful for ensuring online stability. However, excessive regularization may result in slow finetuning. Can we improve existing online off-policy algorithms so tha t online finetuning is more stable without compromising sample efficiency? In the following section, we attempt to answer this question.

## 4 Conservative Policy Improvement in TD3

While offline RL algorithms constrain the policy to be close to the empirical data distribution, such a constraint is inadequate for finetuning since it may be too conservative to allow for fast online learning. On the other hand, constraining policy optimization to the vicinity of a historically good policy may be beneficial since it may limit the influence of an inaccurate critic.

Therefore, we propose to improve the online TD3 algorithm by changing the unconstrained policy improvement step to a constrained update that penalizes large policy updates. Concretely, we propose TD3-C, an off-policy deep RL algorithm based on TD3 that uses the following constrained policy improvement step in place of the original TD3 policy optimization step:

$$\max_{\theta} \quad \mathbb{E}_{s \sim \mathcal{B}}[Q_\phi(s,a)|_{a=\pi_\theta(s)}] \qquad \text{s.t.} \quad \mathbb{E}_{s \sim \mathcal{B}}[(\pi_\theta(s) - \pi_{\theta'}(s))^2] \leq \epsilon. \tag{4}$$

Here $\theta$ is the online policy network parameter, $\theta'$ is the target policy network parameter, and $\epsilon$ is a hyper-parameter that controls the tightness of the constraint. The constraint regularizes the online policy to not deviate too much from a moving target policy. This formulation resembles the constrained optimization used in MPO (Abdolmaleki et al., 2018), except that we work with a deterministic policy and use the $\ell_2$ norm as the constraint. The constraint has the interpretation of a KL-divergence between two Gaussian policies with fixed scales and mean parameterized with the output from the online and target policies' output. For a practical implementation, we can optimize the objective by formulating the Lagrangian with dual variables $\lambda \geq 0$. The constrained optimization now becomes

$$\max_{\theta} \min_{\lambda \geq 0} \quad \mathbb{E}_{s \sim B}[Q_\phi(s,a) - \lambda[\epsilon - (a - \pi_{\theta'}(s))^2]], \quad a = \pi_\theta(s), \tag{5}$$

where the primal $\theta$ and dual variables $\lambda$ can be jointly optimized by stochastic gradient descent.

Unlike the behavior cloning regularization in Equation (3), which constrains the policy network to produce actions similar to those in the sampled batches, we constrain the policy optimization not to take large steps. Thus, our formulation should be less conservative than TD3-BC but more robust than TD3 without policy regularization.

## 5 Evaluation

In this section, we position our results in the context of recent literature that also considers finetuning from offline pretrained agents. We compare our results with Online Decision Transformer (ODT) (Zheng et al., 2022) and Implicit Q-learning (IQL). ODT is a recent approach based on the Decision Transformer (DT) (Chen et al., 2021). Similar to DT, it formulates RL as a sequence modeling problem and leverages the expressiveness of transformer architectures to learn in the RL setting. IQL is an offline Q-learning method that learns the optimal Q-function using in-sample transitions and extracts a policy with advantage weighted regression (AWR) (Neumann and Peters, 2008; Peng et al., 2019). Both approaches have demonstrated better performance when used in the finetuning setting compared to previous work (Nair et al., 2021; Kumar et al., 2020; Fujimoto et al., 2019). For a fairer comparison with IQL and ODT, we consider the setting where we initialized the online replay buffer with the offline dataset. For all methods, we report the offline performance, performance after 200K steps of online interactions and the relative performance improvement $\delta$, computed as the difference between the online and offline performance.

Table 1 shows that our proposed approaches are consistently better than ODT and IQL. This is evident in the better evaluation results after finetuning for 200K steps. Note that a better offline performance cannot explain the better final finetuning performance before finetuning begins since offline learning with TD3-BC

| Task | Agent | medium-replay-v2 | | | medium-v2 | | |
|------|-------|--------|--------|--------|--------|--------|--------|
| | | Offline | Online | $\delta$ | Offline | Online | $\delta$ |
| ant | TD3 | $78.34 \pm 19.5$ | $\mathbf{127.59 \pm 5.24}$ | 49.25 | $\mathbf{113.48 \pm 4.74}$ | $\mathbf{123.34 \pm 2.67}$ | 9.86 |
| | TD3-C | $85.68 \pm 13.93$ | $\mathbf{121.72 \pm 10.06}$ | 36.04 | $\mathbf{112.44 \pm 5.71}$ | $\mathbf{123.05 \pm 2.31}$ | 10.61 |
| | TD3-BC | $84.99 \pm 15.95$ | $\mathbf{126.73 \pm 3.15}$ | 41.74 | $\mathbf{111.44 \pm 5.79}$ | $\mathbf{120.91 \pm 7.97}$ | 9.47 |
| | ODT | $86.56 \pm 3.26$ | $91.57 \pm 2.73$ | 5.01 | $91.33 \pm 4.13$ | $90.79 \pm 5.8$ | $-0.54$ |
| | IQL | $91.21 \pm 7.27$ | $91.36 \pm 1.47$ | 0.15 | $99.92 \pm 5.86$ | $100.85 \pm 2.02$ | 0.93 |
| halfcheetah | TD3 | $43.82 \pm 0.49$ | $\mathbf{70.13 \pm 3.4}$ | 26.3 | $46.99 \pm 0.4$ | $\mathbf{69.69 \pm 2.57}$ | 22.7 |
| | TD3-C | $43.84 \pm 0.69$ | $\mathbf{66.0 \pm 2.06}$ | 22.16 | $46.92 \pm 0.42$ | $\mathbf{65.85 \pm 2.46}$ | 18.93 |
| | TD3-BC | $43.74 \pm 0.53$ | $48.7 \pm 1.18$ | 4.96 | $46.92 \pm 0.41$ | $48.71 \pm 0.52$ | 1.79 |
| | ODT | $39.99 \pm 0.68$ | $40.42 \pm 1.61$ | 0.43 | $42.72 \pm 0.46$ | $42.16 \pm 1.48$ | $-0.56$ |
| | IQL | $44.1 \pm 1.14$ | $44.14 \pm 0.3$ | 0.04 | $47.37 \pm 0.29$ | $47.41 \pm 0.15$ | 0.04 |
| hopper | TD3 | $46.21 \pm 21.91$ | $\mathbf{103.08 \pm 3.7}$ | 56.87 | $54.95 \pm 3.75$ | $\mathbf{88.59 \pm 28.4}$ | 33.65 |
| | TD3-C | $49.18 \pm 25.6$ | $\mathbf{96.24 \pm 11.3}$ | 47.06 | $56.54 \pm 4.71$ | $\mathbf{87.3 \pm 24.62}$ | 30.77 |
| | TD3-BC | $51.36 \pm 24.94$ | $87.72 \pm 14.96$ | 36.36 | $55.48 \pm 4.69$ | $58.44 \pm 6.72$ | 2.96 |
| | ODT | $\mathbf{86.64 \pm 5.41}$ | $88.89 \pm 6.33$ | 2.25 | $66.95 \pm 3.26$ | $\mathbf{97.54 \pm 2.1}$ | 30.59 |
| | IQL | $\mathbf{92.13 \pm 10.43}$ | $\mathbf{96.23 \pm 4.35}$ | 4.1 | $63.81 \pm 9.15$ | $66.79 \pm 4.07$ | 2.98 |
| walker2d | TD3 | $72.56 \pm 11.37$ | $\mathbf{100.06 \pm 5.74}$ | 27.5 | $80.88 \pm 4.9$ | $82.09 \pm 21.38$ | 1.21 |
| | TD3-C | $74.78 \pm 10.97$ | $\mathbf{97.21 \pm 3.07}$ | 22.44 | $79.34 \pm 5.7$ | $78.97 \pm 21.03$ | $-0.36$ |
| | TD3-BC | $79.12 \pm 7.2$ | $\mathbf{90.26 \pm 2.27}$ | 11.14 | $80.73 \pm 3.72$ | $84.56 \pm 2.09$ | 3.82 |
| | ODT | $68.92 \pm 4.79$ | $76.86 \pm 4.04$ | 7.94 | $72.19 \pm 6.49$ | $76.79 \pm 2.3$ | 4.6 |
| | IQL | $73.67 \pm 6.37$ | $70.55 \pm 5.81$ | $-3.12$ | $79.89 \pm 3.06$ | $80.33 \pm 2.33$ | 0.44 |

Table 1: Results with ODT (Zheng et al., 2022) and IQL (Kostrikov et al., 2022) on the D4RL MuJoCo locomotion tasks. The ODT and IQL results were obtained from (Zheng et al., 2022). Our results are averaged over ten seeds. We report the average final offline performance (Offline), the final online performance after 200K of online steps (Online) and the relative performance improvement $\delta = $ Online $-$ Offline. Results for the method with the best performance and comparable alternavies (within 10% of the best method's mean) are in **bold**.

performs no better than ODT and IQL except for ant-medium-v2. Even in cases where pretraining with TD3-BC has better offline performance than ODT or IQL, we still observe more significant performance improvement as evident from the larger relative improvement. In hopper-medium-replay-v2, although pretraining with TD3-BC performs worse during offline learning, finetuning with TD3 or TD3-C allows us to improve significantly.

Note that ODT, considered a strong baseline for finetuning from offline RL, performs hyperparameter tuning for each task and initializes the replay buffer using only top-performing trajectories from the offline dataset. In contrast, our approach is easy to implement and requires minimal changes to existing algorithms during online finetuning. Furthermore, we use the same hyperparameters for all datasets, and the hyperparameters are chosen to be the same as the default values used in previous work. We also do not change how data are sampled from the replay buffer and perform additional pre-processing to the dataset[1]. We also include results for not initializing with the offline dataset during finetuning in Table C3. In this case, finetuning with TD3 and TD3-C still outperforms finetuning with ODT and IQL significantly. At the same time, the performance of finetuning with TD3-BC also improves due to constraining to only online data collected during finetuning.

**Is conservative policy optimization effective at mitigating policy collapse?** We investigate if incorporating conservative policy optimization helps mitigate policy collapse. Figure 3 shows the finetuning performance between TD3, TD3-BC and TD3-C on the eight datasets with or without initializing the online replay with from the offline dataset. While TD3-C improves training stability in both settings, the effect is more significant when no offline data is used during finetuning. The results illustrate the difference between the constraint used in TD3-C and the behavior cloning regularizer in TD3-BC. For TD3-BC, the policy is regularized towards the empirical behavior policy, which is crucial for minimizing extrapolation errors but may result in slow improvement. For TD3-C, we regularize the policy towards the changing target policy, avoiding

---

[1]The original TD3-BC performs observation normalization, which shows improved performance in some environments. In our implementation, we do not normalize the observations.

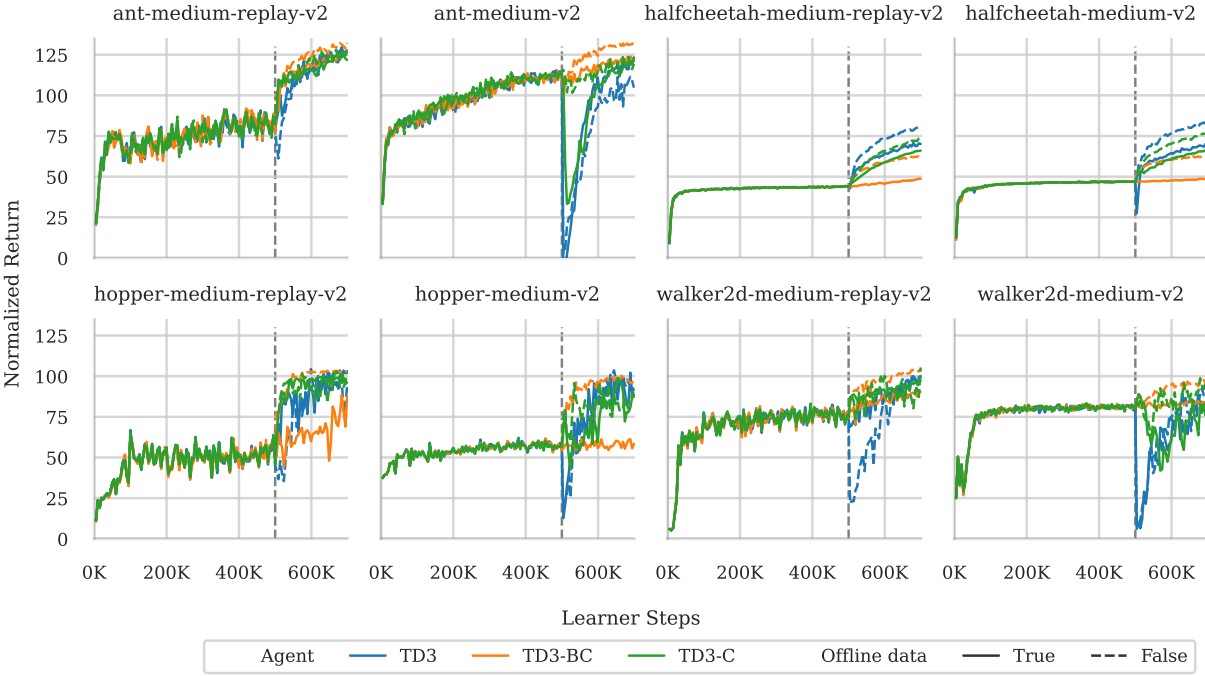

Figure 3: Results of TD3, TD3-BC and TD3-C for finetuning. We compare the three approaches for finetuning varying whether we initialize the online replay buffer with transitions from the offline dataset. TD3-C demonstrates better stability compared to TD3 and can improve faster compared to finetuning with TD3-BC.

large policy changes during optimization due to an inaccurate critic. However, TD3-C can still collapse when preloading the online replay with offline data, as seen in ant-medium-v2. As explained in Section 3.3, the online sampling distribution remains almost unchanged during the initial period of finetuning. Using TD3 or TD3-C on this distribution can suffer from overestimation errors due to the lack of prompt online feedback. Thus, the regularizer in TD3-C, coupled with uniform sampling from an online replay buffer initialized with offline transitions, can still suffer from policy collapse due to optimization with a "fixed" dataset.

## 6 Discussion and Practical Guidelines

Our observations provide a few points for practitioners to consider when leveraging offline RL as a pretraining step and improving offline policies with online data. First, using offline RL methods to continue finetuning may be sub-optimal if the goal is to maximize performance improvement with additional online samples. Using online off-policy algorithms may allow convergence to a better final policy. Second, our empirical results suggest that from a practical perspective, it is crucial to pretrain the agents on datasets that include diverse transitions, provided the offline RL algorithm does not deteriorate significantly with the addition of these suboptimal transitions. While both the policy and the critic can enjoy more robustness and better generalization capability with diverse suboptimal data, the critic should be exposed to both good and bad actions during pretraining to ensure that finetuning with online algorithms is stable.

Our results reveal some limitations on the reported benchmark results in finetuning from offline RL. Previous work (Nair et al., 2021; Kostrikov et al., 2022; Zheng et al., 2022) demonstrates that some offline RL algorithms enjoy better finetuning performance. However, the conclusion is usually made by comparing alternative offline RL methods for finetuning or an online RL baseline. In our paper, we demonstrate that using online RL algorithms with offline RL pretraining is a simple and effective approach that is often not compared with in previous work, such as (Nair et al., 2021; Zheng et al., 2022; Kostrikov et al., 2022) with the exception of (Lee

et al., 2021). However, we do not argue that online RL for finetuning is necessarily superior and should always be preferred. As we have seen in Section 3, policy constraint methods, such as TD3-BC, enjoy better stability, and the performance can sometimes rival online RL algorithms. However, we argue that, instead of utilizing constraints designed for offline learning, there are alternative constraints that would work better for online finetuning. We discussed one approach using conservative policy optimization in Section 4. While we do not find finetuning with TD3-BC to suffer from policy collapse, other works (Lu et al., 2021; Monier et al., 2020) have shown that policy collapse can indeed happen even when using offline RL methods for finetuning. We expect that incorporating conservative policy optimization can also help improve stability in those cases.

Our study also suggests that the evaluation protocols used by some previous works do not sufficiently reflect the challenges we need to address in finetuning from offline RL. We find that the conclusion drawn from evaluating finetuning performance depends on the number of online samples allowed. When a small number of online samples is chosen, the evaluation will favor algorithms with better stability, but not necessarily algorithms that are more finetuning-efficient. In practice, a trade-off between stability and relative performance improvement exists, and we hope future work can also discuss more in more detail during evaluation.

Recent work aiming to improve finetuning from offline RL often incorporates algorithmic improvements (Nair et al., 2021; Kostrikov et al., 2022; Lee et al., 2021; Zheng et al., 2022) coupled with changes to the underlying actor-critic algorithms and replay sampling strategy. This makes it difficult to understand performance improvements coming from individual components, hindering our understanding of what makes finetuning from offline RL difficult. In our paper, we attempt to isolate these changes and demonstrate that, by just changing the online algorithms during finetuning or the online replay initialization, existing algorithms can have a significant performance boost that rivals or outperforms more sophisticated approaches. Given the relative ease of implementing these changes, we hope future work can incorporate them as baselines to measure better our progress on finetuning from offline RL.

## 7 Conclusion

We studied the difficulty in leveraging offline RL as pretraining for online RL. We found that finetuning with offline RL results in slow improvement while finetuning with online RL algorithms is sensitive to distribution shifts. We found that conservative policy optimization is a promising approach for stabilizing finetuning from offline RL when the offline dataset lacks diversity.

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

# Appendix

## A  Broader Impact

Offline RL can enable new applications of reinforcement learning where logged behavior data is available. Examples include robotics, finance and healthcare applications. Our method may have a negative societal impact if deployed in applications that leverage user behavior data to induce addictive behavior in users.

Our contribution considers how to maximize data efficiency of online RL through offline RL pretraining. This can reduce the costs of collecting more online data. However, we do not currently consider the safety aspects of our research. Collecting online data with RL algorithms may be unsafe if the RL algorithms do not explicit take the safety constraints into account.

## B  Implementation Details

Our TD3-C TD3-BC and TD3 implementation are based on the JAX TD3 implementation in Acme (Hoffman et al., 2020). The architectures used by the different algorithms are kept the same: both the policy and the critic network are LayerNormMLPs[2] with 2 hidden layers of size 256 and ELU activation. The LayerNorm layer is inserted after the first linear layer and is followed by tanh activation. The last layer of the policy network is initialized with small weights. This architecture was found to be superior compared to the original architecture used by the TD3 paper for online learning. For simplicity and consistent comparison, we do not perform observation normalization in TD3-BC, which was shown in the original paper to boost performance but unnecessary for achieving good performance. Other hyperparameters are kept the same as in the original TD3 implementation. We found that clipping the gradient of the critic with respect to the policy action improves stability. This is implemented in Acme's DDPG and D4PG implementation but is absent in the original TD3 implementation and the Acme implementation. We clip the gradient from the critic to the policy action to have unit norm in our TD3-C implementation.

Below is a list of hyper-parameters.

Table B1: TD3 and TD3-BC hyper-parameters

| Hyperparameter | Value |
|---|---|
| optimizer | Adam |
| policy learning rate | 3e-4 |
| critic learning rate | 3e-4 |
| target network update rate $\tau$ | 5e-3 |
| delay | 2 |
| maximum replay size | 1e6 |
| minimum replay size | 1000 |
| batch size | 256 |
| exploration noise stddev. | 0.1 |
| target noise stddev. | 0.2 |
| target noise clipping | 0.5 |
| TD3-BC behavior cloning $\alpha$ | 2.5 |

## C  Additional Results

### C.1  Fixing the Ratio of Offline and Online Samples

We investigate the effect of fixing the ratio between offline and online samples during finetuning. The results are illustrated in Figure 4. Finetuning with TD3-BC remains stable and reducing the ratio of offline samples

---

[2]See https://github.com/deepmind/acme/blob/master/acme/agents/jax/td3/networks.py

Table B2: TD3-C hyper-parameters

| Hyperparameter | Value |
| --- | --- |
| $\epsilon$ | 1e-5 |
| clipping | yes |

may allow TD3-BC to improve faster online (e.g., walker2d-medium-v2). Finetuning with TD3 still suffers from policy collapse with varying ratios considered here.

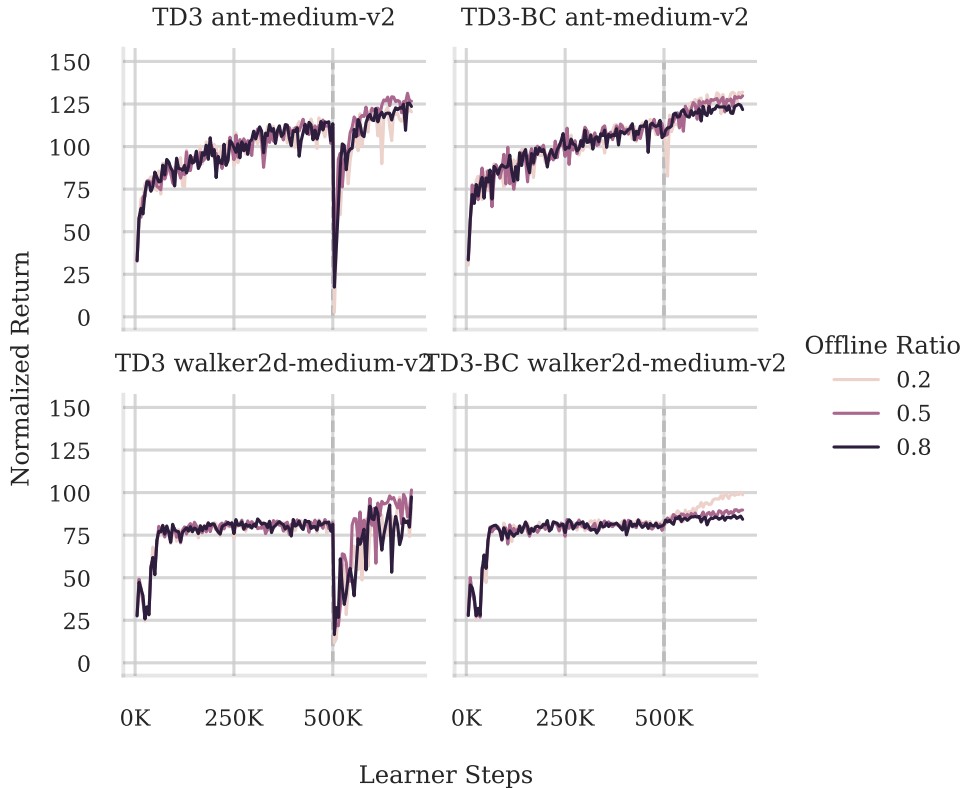

Figure 4: Effect of explicitly limiting the ratio between offline and online samples.

## C.2 Finetuning without Offline Dataset Initialization

Table C3 shows results for finetuning with the TD3, TD3-BC and TD3-C without initializing the online replay buffer with the offline dataset. In this case, we note that performance of TD3-BC improves and trains stably during online finetuning.

| Dataset | Method | Offline | Online | $\delta$ |
|---|---|---|---|---|
| ant-medium-replay-v2 | TD3 | $78.6 \pm 22.55$ | $122.91 \pm 15.16$ | $44.31$ |
| | TD3-C | $82.2 \pm 24.9$ | $125.5 \pm 5.19$ | $43.3$ |
| | TD3-BC | $76.63 \pm 16.31$ | $132.34 \pm 2.92$ | $55.71$ |
| ant-medium-v2 | TD3 | $112.34 \pm 6.11$ | $104.71 \pm 8.48$ | $-7.63$ |
| | TD3-C | $114.15 \pm 4.7$ | $119.34 \pm 7.9$ | $5.19$ |
| | TD3-BC | $112.95 \pm 6.21$ | $132.99 \pm 1.43$ | $20.04$ |
| halfcheetah-medium-replay-v2 | TD3 | $43.77 \pm 0.59$ | $80.98 \pm 3.61$ | $37.21$ |
| | TD3-C | $43.7 \pm 0.63$ | $73.52 \pm 2.17$ | $29.81$ |
| | TD3-BC | $43.83 \pm 0.47$ | $62.35 \pm 1.41$ | $18.52$ |
| halfcheetah-medium-v2 | TD3 | $46.84 \pm 0.43$ | $82.47 \pm 2.7$ | $35.63$ |
| | TD3-C | $46.86 \pm 0.36$ | $75.93 \pm 3.28$ | $29.08$ |
| | TD3-BC | $46.96 \pm 0.37$ | $62.88 \pm 1.25$ | $15.92$ |
| hopper-medium-replay-v2 | TD3 | $44.84 \pm 23.2$ | $94.52 \pm 25.1$ | $49.68$ |
| | TD3-C | $45.69 \pm 22.31$ | $102.38 \pm 3.01$ | $56.69$ |
| | TD3-BC | $49.22 \pm 22.95$ | $102.75 \pm 2.43$ | $53.53$ |
| hopper-medium-v2 | TD3 | $55.14 \pm 4.48$ | $90.91 \pm 33.52$ | $35.78$ |
| | TD3-C | $56.38 \pm 5.86$ | $90.58 \pm 20.91$ | $34.2$ |
| | TD3-BC | $54.35 \pm 4.86$ | $98.8 \pm 9.72$ | $44.45$ |
| walker2d-medium-replay-v2 | TD3 | $72.66 \pm 10.59$ | $98.94 \pm 10.38$ | $26.28$ |
| | TD3-C | $75.24 \pm 9.78$ | $105.35 \pm 11.01$ | $30.12$ |
| | TD3-BC | $73.55 \pm 8.18$ | $104.16 \pm 4.66$ | $30.61$ |
| walker2d-medium-v2 | TD3 | $80.16 \pm 5.1$ | $85.19 \pm 19.55$ | $5.03$ |
| | TD3-C | $80.41 \pm 4.99$ | $97.54 \pm 11.88$ | $17.13$ |
| | TD3-BC | $82.26 \pm 1.32$ | $97.48 \pm 3.45$ | $15.22$ |

Table C3: Results for TD3, TD3-BC and TD3-C without initializing the online replay with the offline dataset.

## C.3 Additional Comparison

### C.3.1 Comparison to AWAC and RLPD

In this section, we compare our results with RLPD (Ball et al., 2023) and AWAC (Nair et al., 2021). TD3, TD3-BC and TD3-C all outperform AWAC in terms of final finetuning performance. We found that using high UTD ratio improves performance by RLPD significantly. However, with a UTD ratio of 1, RLPD underperforms TD3, TD3-BC and TD3-C.

| Method Dataset | AWAC mean | std | RLPD (UTD=1) mean | std | RLPD (UTD=20) mean | std | TD3 mean | std | TD3-BC mean | std | TD3-C mean | std |
|---|---|---|---|---|---|---|---|---|---|---|---|---|
| ant-mr-v2 | 108.57 | 3.18 | 111.27 | 9.08 | 148.98 | 0.95 | 122.91 | 15.16 | 132.34 | 2.92 | 125.50 | 5.19 |
| ant-m-v2 | 110.71 | 1.87 | 122.57 | 12.18 | 152.97 | 0.93 | 104.71 | 8.48 | 132.99 | 1.43 | 119.34 | 7.90 |
| halfcheetah-mr-v2 | 46.16 | 0.59 | 64.12 | 0.76 | 88.20 | 2.90 | 80.98 | 3.61 | 62.35 | 1.41 | 73.52 | 2.17 |
| halfcheetah-m-v2 | 48.07 | 0.30 | 62.54 | 4.78 | 92.13 | 0.77 | 82.47 | 2.70 | 62.88 | 1.25 | 75.93 | 3.28 |
| hopper-mr-v2 | 67.43 | 25.49 | 57.93 | 39.49 | 86.07 | 20.95 | 94.52 | 25.10 | 102.75 | 2.43 | 102.38 | 3.01 |
| hopper-m-v2 | 51.79 | 10.03 | 93.64 | 12.35 | 99.86 | 15.86 | 90.91 | 33.52 | 98.80 | 9.72 | 90.58 | 20.91 |
| walker2d-mr-v2 | 87.49 | 2.44 | 76.38 | 21.34 | 116.46 | 3.16 | 98.94 | 10.38 | 104.16 | 4.66 | 105.35 | 11.01 |
| walker2d-m-v2 | 80.55 | 7.62 | 98.19 | 4.91 | 118.54 | 2.01 | 85.19 | 19.55 | 97.48 | 3.45 | 97.54 | 11.88 |
| locomotion-total-v2 | 600.76 | | 686.64 | | 903.23 | | 760.63 | | 793.75 | | 790.14 | |

Table C4: Comparison to AWAC and RLPD.

### C.3.2 Comparison to Balanced Replay

We also compared with the Balanced Replay approach from (Lee et al., 2021). We include the reported results from (Lee et al., 2021). We also include results from our re-implementation of (Lee et al., 2021) as the implementation from (Lee et al., 2021) does not provide the code for reproducing their offline training. In addition, we include a baseline that uses CQL for offline training and SAC for online finetuning (without offline data) to isolate the effect of the balanced-replay introduced in (Lee et al., 2021). We found that finetuning with offline algorithms such as TD3-BC but removing any offline data during finetuning allows us to mostly achieve similar performance compared to finetuning with online algorithms (SAC) with offline RL initialization for many environments except for halfcheetah where we found that any type of policy constraint hurts finetuning sample efficiency.

| Dataset | CQL->SAC | Off2On (Ours) | Off2On (Lee et al., 2021) | TD3-BC |
|---|---|---|---|---|
| ant-m-v2 | 130.6±8.62 | 131.7±13.5 | | 132.99±1.43 |
| ant-mr-v2 | 117.8±11.7 | 127.6±10.5 | | 132.34±2.92 |
| halfcheetah-m-v2 | 93.3±2.79 | 93.7±3.2 | ≈ 9000 (74.4) | 62.88±1.25 |
| halfcheetah-mr-v2 | 84.5±1.82 | 85.5±2.4 | ≈ 8000 (66.7) | 62.35±1.41 |
| hopper-m-v2 | 92.7±34.8 | 98.7±18.5 | ≈ 3300 (102.0) | 98.8±9.72 |
| hopper-mr-v2 | 97.7±17.2 | 91.8±21.8 | ≈ 3400 (105.1) | 102.75±2.43 |
| walker2d-m-v2 | 83.2±20.4 | 95.8±15.8 | ≈ 4500 (98.0) | 97.48±3.45 |
| walker2d-mr-v2 | 88.9±24.2 | 113.1±4.6 | ≈ 4500 (98.0) | 104.16±4.66 |
| locomotion-total-v2 | 788.7 | 837.9 | | 793.75 |

Table C5: Comparison to Off2On

## C.4 Different choices of offline/online RL algorithms

We investigate if our findings on the effect of online data composition generalize to other RL algorithms. In particular, we consider using CQL for both offline pretraining (Kumar et al., 2020) and online finetuning but

vary how data is sampled during online finetuning similar to the investigation we performed in Section 3.3. The results are summarized in Table C6. Similar to our findings for TD3-BC, online finetuning with offline algorithms such as CQL improve slowly but the performance can be significantly improved by discarding offline data during online finetuning.

| Dataset | CQL->CQL(with offline data) | CQL->CQL(online only) |
|---|---|---|
| ant-m-v2 | 109.3 ± 4.93 | 117.3±3.04 |
| ant-mr-v2 | 108.7 ± 5.63 | 111.2±4.30 |
| halfcheetah-m-v2 | 50.95±0.47 | 71.1±2.41 |
| halfcheetah-mr-v2 | 51.56±1.74 | 65.5±4.68 |
| hopper-m-v2 | 74.7±6.73 | 98.7 ± 4.50 |
| hopper-mr-v2 | 98.3±2.63 | 92.7 ± 13.0 |
| walker2d-m-v2 | 82.8±1.79 | 92.8 ± 6.53 |
| walker2d-mr-v2 | 87.7±4.44 | 96.7 ± 5.89 |
| locomotion-total-v2 | 664.0 | 746.0 |

Table C6: Effect of online data buffer for CQL

We also consider equipping another offline RL algorithm, IQL (Kostrikov et al., 2022), with different ratios of online data during online training. Figure 5 shows the results for IQL finetuning on antmaze with different ratios of online data. The result is consistent with our findings with TD3-BC: when finetuning with offline RL algorithms, changing the ratio of online data provide benefits for improving the online finetuning efficiency without sacrificing stability. This is true for all antmaze datasets we consider here except antmaze-umaze-diverse-v2 where we found that IQL diverges after online finetuning.

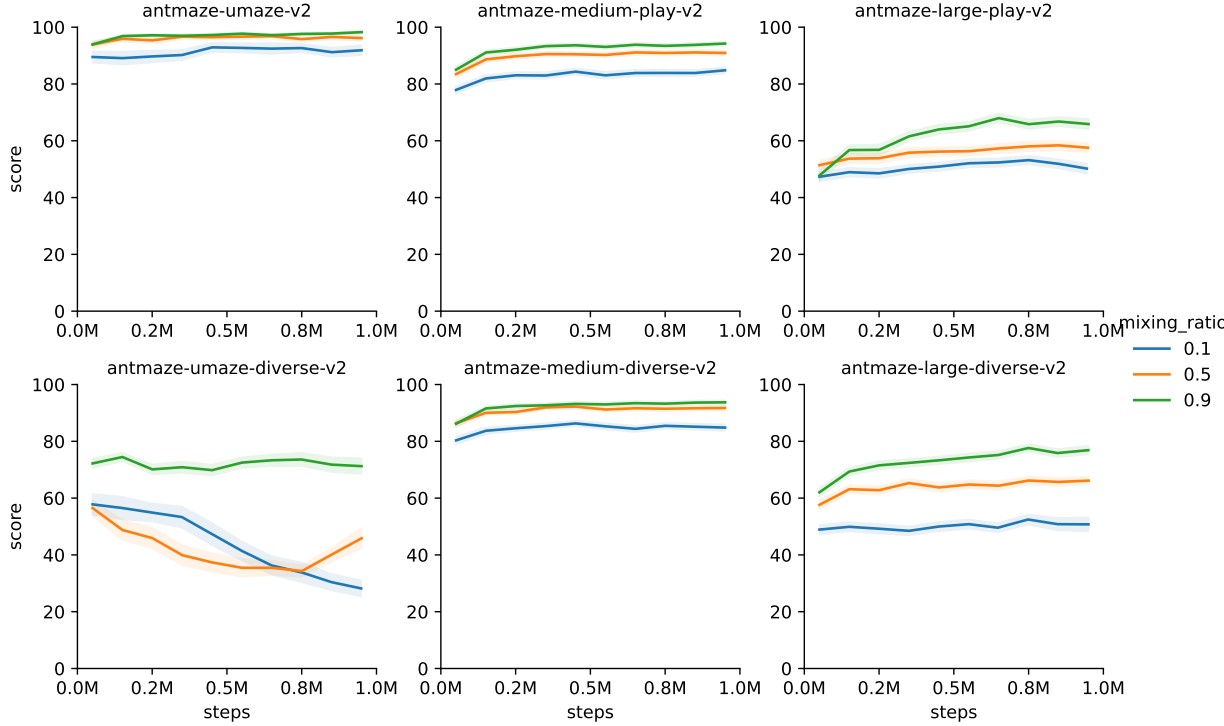

Figure 5

## C.5 Ablation on TD3 hyperparameters

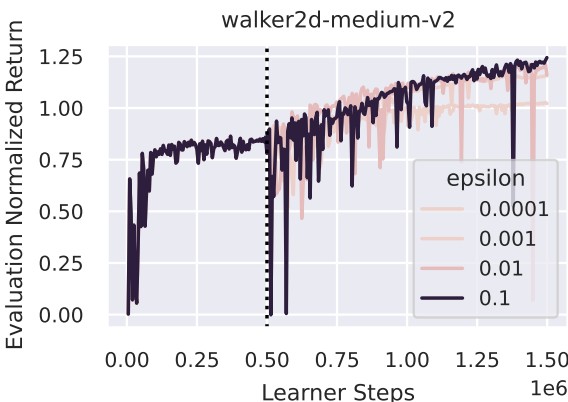

Figure 6: Effect of varying $\epsilon$. Smaller $\epsilon$ leads to more stable online finetuning but slower improvement.

**Improving training stability with more conservative policy improvement.** TD3-C introduces an $\epsilon$ parameter that determines the level of the constraint used during policy improvement. We investigate different choices of $\epsilon$. The results are shown in Figure 6. Generally, we find that choosing stronger constraints lead to more stable finetuning but may result in slower finetuning efficiency.

**Improving training stability with more delayed policy improvements.** The policy improvement constraint penalty stabilizes learning for the medium datasets. However, when only expert demonstrations are used in offline learning, the critic suffers from severe overfitting, and constrained updates alone are insufficient to prevent policy collapse. In this case, we found that updating the critic more frequently stabilizes finetuning. Note that reducing update frequency is different from constraining policy updates: constraining the policy optimization has the added benefit of making policy optimization more robust to critic error. Our concrete implementation takes advantage of the delay parameter in TD3, which determines the frequency of policy optimization. Figure 7 compares different delayed step values with evaluation performance. We see that increasing the delay helps prevent policy collapse during initial finetuning. This ablation suggests that when adopting standard off-policy algorithms for finetuning, reducing the ratio of policy improvement to policy evaluation steps may help.

## C.6 Negative Results

The critic suffers from significant extrapolation error when first deployed online. Therefore, we considered many approaches that we thought will mitigate this issue prior to the approach that we will discuss in Section 4. However, we have not been able to succeed with any of these ideas. This section shows that bootstrap error even under mild distribution shift is severe and that standard regularization is insufficient to ensure stable online finetuning.

We tried running the policy evaluation only in the first 500K online steps before improving the policy and still observed significant policy collapse as soon as we start updating the policy. This suggests that it is not sufficient to avoid collapse by having a more accurate critic with online policy evaluation. In fact, we found that in many cases, the training loss for the critic during finetuning is similar to that during offline learning when the policy is not further updated online. This suggests that the critic is at least accurate at evaluating the policy. However, policy collapse happens immediately as soon as we start running policy optimization during finetuning. Thus the policy collapse comes from the fact that critic becomes severely inaccurate even with mild chnages in policy. We tried regularizing the policy and critic optimization with weight decay, clipping the gradients in the policy and/or the critic, using a larger critic network but none

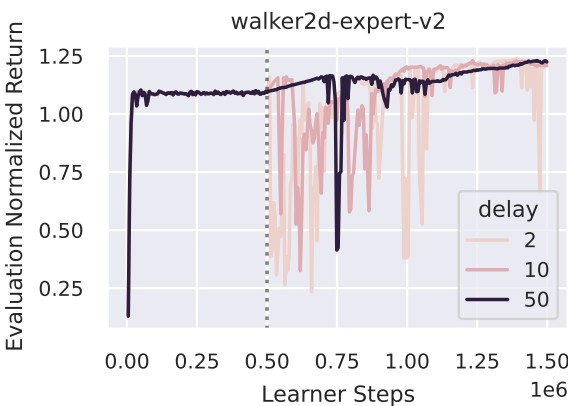

Figure 7: Effect of increasing the number of delay steps in TD3-C. Constrained updates alone do not prevent policy collapse, but reducing the amount of policy improvement steps mitigates it.

seems to mitigate this issue. The policy collapse suggests that the offline trained critic overfits, so we tried improving generalization with data augmentation (Laskin et al., 2020), but this is ineffective for preventing policy collapse. We tried a distributional critic or using an ensemble of critic networks, i.e., instead of using two critics as in TD3, we trained ten critics in parallel in the hope of further reducing overestimation bias. We also tried using uncertainty in the critic ensemble to penalize policy improvements, but we have not managed to take advantage of it to prevent policy collapse.

