# OpenReview forum: "Finetuning from Offline Reinforcement Learning: Challenges, Trade-offs and Practical Solutions"
_TMLR — Rejected by TMLR_

### Review · Reviewer_BH3h · 2023-06-26

**Summary Of Contributions:**

This paper explores how to improve the performance of pre-trained policies in offline reinforcement learning through fine-tuning. It is found that using standard online off-policy algorithms for fine-tuning can lead to faster improvement, but it may also result in policy collapse. To address this issue, a conservative policy optimization method is proposed by analyzing factors such as data diversity, algorithm selection, and online replay distribution. This method achieves stable and sample-efficient online learning.

**Audience:**

Yes

**Claims And Evidence:**

Yes

**Requested Changes:**

1. Based on the content of the manuscript, it appears that the unconstrained policy improvement could potentially be applied with various offline RL algorithms for offline-to-online learning. However, the experiments conducted in the current work only utilize TD3. It would be beneficial if the authors could validate its effectiveness by incorporating other offline RL algorithms such as CQL[1] or IQL[2].
2. The focus of this paper is on the offline-to-online RL problem. However, in the experimental section, only one related method, ODT, is compared, while other well-known approaches such as AWAC[3] and Balanced Replay[4] are not included for comparison. It is essential for the authors to compare their proposed method against these prominent approaches in the experimental section to establish the effectiveness of their proposed method.
3. Based on the experimental results, it is observed that TD3-C does not effectively address the performance drop issue. The online phase exhibits significant fluctuations on certain datasets. This outcome suggests that solely employing policy update magnitude constraints may have limitations. Therefore, it is recommended that the authors consider incorporating additional mechanisms to achieve improved performance.
4. For the offline-to-online RL setting, using line plots instead of tables would better illustrate the differences between various methods. It is recommended that the authors utilize line plots more frequently to compare the baselines. Additionally, the design of the line plots should be carefully considered. The current line plots appear somewhat disorganized and could benefit from improved clarity and organization.
5. In recent years, there has been rapid development in the field of offline-to-online RL, with numerous relevant works published. However, the background section of the manuscript only references early works, overlooking several newer contributions. It is crucial for the authors to include and discuss these more recent works in the related works section, such as PEX[5], E2O[6], and PROTO[7].

[1] Kumar A, Zhou A, Tucker G, et al. Conservative q-learning for offline reinforcement learning[J]. Advances in Neural Information Processing Systems, 2020, 33: 1179-1191.

[2] Kostrikov I, Nair A, Levine S. Offline reinforcement learning with implicit q-learning[J]. arXiv preprint arXiv:2110.06169, 2021.

[3] Nair A, Gupta A, Dalal M, et al. Awac: Accelerating online reinforcement learning with offline datasets[J]. arXiv preprint arXiv:2006.09359, 2020.

[4] Lee S, Seo Y, Lee K, et al. Offline-to-online reinforcement learning via balanced replay and pessimistic q-ensemble[C]//Conference on Robot Learning. PMLR, 2022: 1702-1712.

[5] Zhang H, Xu W, Yu H. Policy Expansion for Bridging Offline-to-Online Reinforcement Learning[J]. arXiv preprint arXiv:2302.00935, 2023.

[6] Zhao K, Ma Y, Liu J, et al. Ensemble-based Offline-to-Online Reinforcement Learning: From Pessimistic Learning to Optimistic Exploration[J]. arXiv preprint arXiv:2306.06871, 2023.

[7] Li J, Hu X, Xu H, et al. PROTO: Iterative Policy Regularized Offline-to-Online Reinforcement Learning[J]. arXiv preprint arXiv:2305.15669, 2023.


**Strengths And Weaknesses:**

1. The investigation of the offline-to-online problem in this work is highly relevant and necessary for practical applications, aligning well with the requirements of real-world scenarios.
2. The proposed method in this paper demonstrates a reasonable approach, utilizing constraints on the magnitude of policy updates during the online phase to achieve stable online optimization. This strategy is not only convenient to implement but also yields promising results.

---

> ### Author Response · Authors · 2023-07-25
> **Reply to Reviewer BH3h**
>
> Dear Reviewer Curu, thank you for helping us improve our manuscript.
>
> ----
> > Based on the content of the manuscript, it appears that the unconstrained policy improvement could potentially be applied with various offline RL algorithms for offline-to-online learning. However, the experiments conducted in the current work only utilize TD3. It would be beneficial if the authors could validate its effectiveness by incorporating other offline RL algorithms such as CQL[1] or IQL[2].
>
> Thank you, we added additional results on CQL and IQL. Hopefully those would be useful for demonstrating the generality of the points we have in this paper.
>
> > The focus of this paper is on the offline-to-online RL problem. However, in the experimental section, only one related method, ODT, is compared, while other well-known approaches such as AWAC[3] and Balanced Replay[4] are not included for comparison. It is essential for the authors to compare their proposed method against these prominent approaches in the experimental section to establish the effectiveness of their proposed method.
>
> We added more baseline results to the appendix. We will update the main manuscript to include these results soon.
>
> > Based on the experimental results, it is observed that TD3-C does not effectively address the performance drop issue. The online phase exhibits significant fluctuations on certain datasets. This outcome suggests that solely employing policy update magnitude constraints may have limitations. Therefore, it is recommended that the authors consider incorporating additional mechanisms to achieve improved performance.
>
> Thank you, we discussed additional things we found possible to further improve performance in the appendix, as well as negative results which we hope would be useful to the community.
> Empirically, we found that TD3-C and TD3 still underperform offline algorithms in terms of stability compared to just using offline RL algorithms.
>
> -----
>
> > For the offline-to-online RL setting, using line plots instead of tables would better illustrate the differences between various methods. It is recommended that the authors utilize line plots more frequently to compare the baselines. Additionally, the design of the line plots should be carefully considered. The current line plots appear somewhat disorganized and could benefit from improved clarity and organization.
>
> Thank you. We will include the baselines in the lineplots. We are currently implementing all baselines in a consistent framework and open-source the implementation to help future work, but this is still WIP. For the final version, we intend to include lineplots for more baselines we have considered in this work.
>
> -----
>
> > In recent years, there has been rapid development in the field of offline-to-online RL, with numerous relevant works published. However, the background section of the manuscript only references early works, overlooking several newer contributions. It is crucial for the authors to include and discuss these more recent works in the related works section, such as PEX[5], E2O[6], and PROTO[7].
>
> Thank you, we were aware that there has been recent work that is concurrent or later than ours. We will discuss more related work covering PEX, E2O, PROTO as well Cal-QL, RLPD more extensively in the final verison. We didn’t manage to do it right now as adding additional baseline results takes a significant portion of the time.

---

### Review · Reviewer_CUru · 2023-06-27

**Summary Of Contributions:**

This paper studies the offline-to-online RL setting. It first presents experiments on the mujoco tasks from the D4RL benchmark demonstrating policy collapse when simple methods are used (e.g. combinations of TD3+BC and TD3 with and without using the offline data as the initial replay buffer). Then the paper proposes and evaluates a slight variant of TD3 called TD3-C and evaluates the method on the same benchmark.

**Audience:**

No

**Claims And Evidence:**

No

**Requested Changes:**

To improve the paper, all of the weaknesses above need to be addressed. Most important is comparison to more recent related work. Second most important is to present more convincing evidence for the proposed TD3-C method. The other issues are relatively minor, but should also be addressed.

**Strengths And Weaknesses:**

Strengths:

1. The paper does a good job of presenting a variety of experiments using consistent and simple methodology. Section 3 presents a variety of ablations of combining TD3 and TD3+BC that (to my knowledge) do not quite exist in prior work, but are potentially helpful to the community by providing simple baseline results of various algorithmic choices.


Weaknesses:

1. The most major weakness is a lack of comparison or discussion of recent work on offline-to-online RL. For example [1] and [2] have been out for several months and report much stronger results than the baselines considered in this paper (and on more challenging domains from D4RL, e.g. antmaze). THey also both take on essentially the same issue of policy collapse and develop more thoroughly motivated solutions (both in theory and practice) than TD3-C.

2. It is not clear from the results in the paper that TD3-C is actually better than the baselines. For example, in table 1 it seems that TD3 is outperforming TD3-C. And figure 3 does not provide any very clear takeaway that TD3-C is the best option either. Moreover, there is no clear theoretical motivation for the algorithm, so these empirical results are the only real evidence in support of the new method.

3. Some of the claims in the paper are unclear or unsubstantiated. For example, there are repeated references to improved performance when pretraining on "diverse" transitions. What is meant by "diverse" here? Does it just mean that you use a "replay" dataset rather than "medium" or "expert"? If so, that evidence does not seem sufficient to make a more general claim beyond the specific datasets that were tested here.

4. Some of the experimental methodology could be cleaned up. For example, why do figures 2 and 3 not have error bars? Why is the replay ration set to 1 for online learning even when recent work has often found that performing more updates per environment step can be beneficial?

[1] Ball, Philip J., Laura Smith, Ilya Kostrikov, and Sergey Levine. "Efficient online reinforcement learning with offline data." arXiv preprint arXiv:2302.02948 (2023).

[2] Nakamoto, Mitsuhiko, Yuexiang Zhai, Anikait Singh, Max Sobol Mark, Yi Ma, Chelsea Finn, Aviral Kumar, and Sergey Levine. "Cal-ql: Calibrated offline rl pre-training for efficient online fine-tuning." arXiv preprint arXiv:2303.05479 (2023).

---

> ### Author Response · Authors · 2023-07-25
> **Reply to Reviewer Curu**
>
> Dear Reviewer Curu, we would like to thank you for helping us improving our manuscript.
>
> ----
>
> > The paper does a good job of presenting a variety of experiments using consistent and simple methodology. Section 3 presents a variety of ablations of combining TD3 and TD3+BC that (to my knowledge) do not quite exist in prior work, but are potentially helpful to the community by providing simple baseline results of various algorithmic choices.
>
> Thank you. We really appreciate that you value our consistent and simple methodology. We believe doing so allows the readers to see straight to the point.
>
> ----
>
> > The most major weakness is a lack of comparison or discussion of recent work on offline-to-online RL. For example [1] and [2] have been out for several months and report much stronger results than the baselines considered in this paper (and on more challenging domains from D4RL, e.g. antmaze). THey also both take on essentially the same issue of policy collapse and develop more thoroughly motivated solutions (both in theory and practice) than TD3-C.
>
> Thank you, yes indeed some of the issues that we discuss in this paper are also discussed concurrent work namely [1] and [2]. We argue, however, that the analysis we performed here is more finegrained and hope our paper can provide a useful reference.
>
> We do not prescribe the approach discussed in this paper to be state-of-the-art. We choose TD3-BC as the main algorithm used in this paper mainly for its simplicity and for its straightforward online RL counterparts. Early in our experiments we tried a few combinations of offline/online algorithms and found that the conclusion we described for TD3-BC/TD3 is applies to other algorithms as well.
>
> ----
>
> > Some of the claims in the paper are unclear or unsubstantiated. For example, there are repeated references to improved performance when pretraining on "diverse" transitions. What is meant by "diverse" here? Does it just mean that you use a "replay" dataset rather than "medium" or "expert"? If so, that evidence does not seem sufficient to make a more general claim beyond the specific datasets that were tested here.
>
> Sorry for the lack of clear definition for “diverse” datasets here. Comparing medium-replay with the medium dataset, the medium-replay dataset includes low-performing trajectories collected early during policy learning. We hypothesize that including these low-performing negative examples in offline RL allows the algorithms to learn more robustly as they can learn a more accurate critic on a state for the different actions performed.
>
> ----
>
> > Some of the experimental methodology could be cleaned up. For example, why do figures 2 and 3 not have error bars? Why is the replay ration set to 1 for online learning even when recent work has often found that performing more updates per environment step can be beneficial?
>
> We omitted the error bar for the figure as it was cluttering with the observing the policy collapse. We will incorporate the error bars in the final version.
>
> We also investigated using a fixed replay ratio similar to those used in [1,2] in the appendix, we found that using a higher online ratio allows faster finetuning using TD3-BC but policy collapse is a consistent problem for TD3 regardless of the online data composition.
>
> -----
>
> [1] Ball, Philip J., Laura Smith, Ilya Kostrikov, and Sergey Levine. "Efficient online reinforcement learning with offline data." arXiv preprint arXiv:2302.02948 (2023).
>
> [2] Nakamoto, Mitsuhiko, Yuexiang Zhai, Anikait Singh, Max Sobol Mark, Yi Ma, Chelsea Finn, Aviral Kumar, and Sergey Levine. "Cal-ql: Calibrated offline rl pre-training for efficient online fine-tuning." arXiv preprint arXiv:2303.05479 (2023).

---

### Review · Reviewer_Gbrz · 2023-07-14

**Summary Of Contributions:**

This paper propounds a pivotal question: In what optimal manner can we refine the agents obtained from offline Reinforcement Learning (RL) methods utilizing an offline dataset? The author further puts forth that the primary challenges associated with transitioning from offline to online RL are twofold: In theory, offline RL is suitable for fine-tuning, yet the rate of improvement in online performance is disappointingly slow. Conventional off-policy RL methods, while capable of driving faster progress, are nonetheless susceptible to policy collapse during the initial stages of online implementation. This treatise investigates the predicament of policy collapse, examining its association with variables such as data diversity, the selection of algorithms, and the distribution of online replays. With these observations in mind, the paper recommends a conservative policy optimization procedure, designed to facilitate stable and highly sample-efficient online learning in the aftermath of offline pre-training.

**Audience:**

Yes

**Claims And Evidence:**

Yes

**Requested Changes:**

1. The section "Summary of Empirical Observations" should be reorganized in a better way. The content of this section should be easier for readers to understand.
2. The performance comparison presented in Table 1 should be replaced by a comparison of performance curves in the form of figure, as in online training, the stability of performance is also a criterion for distinguishing between good and bad algorithms.
3. The focus of offline-to-online RL is on the online learning stage. More timesteps should be used to present performance changes during the online learning phase, rather than offline (For example, in Figure 3). The performance of offline pretrained policy can be simply represented by a straight line.
4. The Medium-Expert dataset is also an important one, it should also be used for experiments.


**Strengths And Weaknesses:**

* Strengths

This paper undertakes a comprehensive analysis of why offline Reinforcement Learning (RL) or off-policy RL methodologies cannot be straightforwardly implemented for offline-to-online RL transition. The author meticulously delineates the concept of policy collapse, articulating the reasons for its occurrence with clarity. This paper offers profound insights into the complex challenges associated with the transition from offline to online RL. Several influential factors impacting the performance of offline-to-online RL have been empirically scrutinized, namely, the choice of online algorithms, the diversity inherent in the dataset, and the level of conservatism inherent in RL algorithms.

* Weaknesses

1. The empirical observations garnered during the experimental phase of this study appear to be cursorily organized, engendering an impression that the paper merely delineates the experimental outcomes.

2. The methodology put forth in this article to address the challenges associated with the transition from offline to online RL is lacking in novelty. The optimization objectives outlined in Equation 4 bear significant resemblance to those in AWAC, with the only discernible difference being the replacement of the behavior policy with the target policy.

3. The author neglected to provide a comparative figure detailing the online performance curves of the state-of-the-art methods and TD3-C. The performance curve of the offline-to-online RL methods tends to fluctuate considerably, thus the data in Table 1 does not adequately reflect the quality of the proposed method.

4. Figure 3 suggests that the performance of TD3-C may not be adequately robust. More online time steps are necessitated to convincingly demonstrate the convergence of TD3-C, particularly as, when utilizing the Medium dataset, the performance curves of the TD3-C were characterized by consistent fluctuations.

5. The performance metrics of AWAC and BRPQ methods should continue to serve as points of reference.

---

> ### Author Response · Authors · 2023-07-25
> **Reply to Reviewer Gbrz**
>
> Dear Reviewer Gbrz, thank you for spending time providing feedback for improving our manuscript.
>
> ----
>
> > The methodology put forth in this article to address the challenges associated with the transition from offline to online RL is lacking in novelty. The optimization objectives outlined in Equation 4 bear significant resemblance to those in AWAC, with the only discernible difference being the replacement of the behavior policy with the target policy.
>
> We do not claim algorithmic novelty in our paper. The objective we used in Eqn 4 is quite common in many more recent online algorithms (The idea of regularizing the updating policy with a target policy to avoid large policy updates which is used in algorithms such as MPO). Our introduction of TD3-C mainly served to illustrate that a commonly used online policy-constraint approach can be useful in addressing policy collapse issues.
>
> ----
>
> > The author neglected to provide a comparative figure detailing the online performance curves of the state-of-the-art methods and TD3-C. The performance curve of the offline-to-online RL methods tends to fluctuate considerably, thus the data in Table 1 does not adequately reflect the quality of the proposed method.
>
> Thank you, we do not intend our approach to outperform state-of-the-art. Instead, we hope our very controlled experiments allow us to derive some insights on simple algorithmic choices.
>
> ----
>
> > Figure 3 suggests that the performance of TD3-C may not be adequately robust. More online time steps are necessitated to convincingly demonstrate the convergence of TD3-C, particularly as, when utilizing the Medium dataset, the performance curves of the TD3-C were characterized by consistent fluctuations.
>
> Thank you. We found the constraint parameter to be important for TD3-C’s robustness.
> We added additional ablation to the appendix. It’s possible to further improve robustness of TD3-C by carefully tuning the epsilon parameter depending on dataset and environment but we do not think it’s helpful for this work.
>
> ----
>
> > The performance metrics of AWAC and BRPQ methods should continue to serve as points of reference.
>
> Thank you, we added results for AWAC and BRPQ in the appendix
>
> ----
>
> > The focus of offline-to-online RL is on the online learning stage. More timesteps should be used to present performance changes during the online learning phase, rather than offline (For example, in Figure 3). The performance of offline pretrained policy can be simply represented by a straight line. The Medium-Expert dataset is also an important one, it should also be used for experiments.
>
> Thank you, we omitted the medium-expert dataset as TD3-BC can reach expert-level performance on these datasets; so it is more difficult to observe improvements during the online stage.

---

### Author Response · Authors · 2023-07-25
**Reply from the authors.**

We would like to thank all reviewers for their feedback and comments.

## Overall replies for all reviewers:
We present an empirical investigation of the problem of online finetuning from offline RL.
We study two important ingredients in finetuning from offline RL:

1. The choice of online algorithms for finetuning.
2. How data is sampled during finetuning.

Previous and concurrent work have found that these components are crucial for finetuning stability and efficiency, yet their individual effects have sometimes been buried deep in other algorithmic changes. Our primary contribution is to disentangle the effects of these two ingredients on finetuning stability and sample efficiency. Ablating carefully over the design choices, we observe that variations along the two axes above can significantly improve finetuning with just offline RL algorithms.
We believe these results are important since otherwise advancements in finetuning from offline RL due to algorithmic improvement may be misinterpreted. We do not intend our work to advance the state of the art in finetuning from offline RL. However, we demonstrate approaches that are (unlike many others) simple to implement and evaluate. As a result, our paper contributes to scientific knowledge and understanding, while it can be a useful baseline for future work.
We choose to use TD3-BC and TD3 to drive the main narratives of our manuscript. This allows us to provide focused discussion on choices of online algorithm and choice of replaying online data. While it’s possible to consider a different offline/online algorithm pair, other choices may be less straightforward than it first appears. For example:
1. The choice of CQL -> SAC is not straightforward. While it’s true that CQL introduces an additional critic regularization term to the SAC objective, in practice, working implementations of CQL frequently require significant changes to SAC, such as network architecture, whether to back up entropy and how to compute targets for policy evaluation, which makes it substantially different from a vanilla SAC implementation.
2. Using IQL as a backbone. There isn’t a straightforward online counterpart for IQL. We did consider using SAC for finetuning IQL checkpoints, but the network’s architecture used by IQL also differs from SAC mostly around the use of policy that is not tanh-squashed and has state-independent scale.

Among the choices that we considered, we find TD3/TD3-BC to be the most straightward for the narrative as the only change in TD3-BC to TD3 is the introduction of an additional BC term. However, note that neither TD3-BC nor TD3 represent state-of-the-art algorithms for offline/online RL. As a result, we are mostly limited to doing a thorough analysis on the locomotion datasets as TD3-BC performs poorly on more challenging tasks such as antmaze and adroit. Nevertheless, we added additional experiments to provide empirical evidence that our results generalize to other algorithms (see updated appendix).

Many reviewers have expressed their concerns on the proposed approach TD3-C that produces results that fluctuate considerably. Our TD3-C experiments are mainly used to demonstrate that employing policy constraints commonly in online RL algorithms can be useful in mitigating policy collapse, although these policy constraints seem to be less effective in general compared to constraints used by pure offline RL algorithms, as shown in the results of our paper.

To be critical about TD3-C, we found that TD3-C provides more improvements in the case where offline data is used to initialize the online replay. We included additional results in the appendix where we omit any offline data during online finetuning. In that case, we observe significant improvements in TD3-BC’s finetuning performance, and in many cases it rivals TD3-C with more finetuning stability. This demonstrates that adjusting the level of online data ratio can significantly improve the rate of finetuning for offline RL algorithms. We think this result is practically useful for the community and provides a simple setup that is frequently omitted in the evaluation section from previous work.

---

> ### Author Response · Authors · 2023-07-25
> **Comparison to BRPQ and AWAC**
>
> We include three additional results for BRPQ and AWAC. CQL->SAC refers to offline learning with CQL and finetuning with SAC. Our Balanced Replay is a reimplementation of the balanced replay from [1] without ensembles. We also adapted approximate results of the original balanced replay paper from Figure 3 of BRPQ. Our reimplementation performs on par or better than the adapted results.   We would like to draw reviewers’ attention to our results for offline TD3-BC and online TD3-BC using online data only (in Table C3). This baseline closely matches the balanced replay results in most of the datasets we considered. The only exception is halfcheetah where we found finetuning with TD3 works best and does not suffer from policy collapse (online TD3 achieved 81 for halfcheetah-mr and 82 for halfcheetah-m). Still, TD3 falls behind CQL->SAC without the balanced replay. This suggests that the difference in the result is due to our different combination of offline/online algorithms, rather than the introduction of balanced replay. We find that using only the only data is sufficient for us to improve the finetuning efficiency of a policy-constraint offline RL algorithm to match state-of-the-art results. We think BRPQ is a valuable paper that incorporates many important ideas, which we advocate in our paper, but at the same time, we show that there are minimalistic approaches that can immediately improve sample efficiency for their finetuning experiments.
>
> We believe it’s important to understand how our simple setup is rarely compared to and hope that our paper should bring more visibility to simple approaches like ours. Previous work that considers finetuning from offline RL:
> 1. Finds that certain types of constraints used in offline RL algorithms make them difficult to finetune [2]. These works compare with using alternative offline RL algorithms for finetuning but did not carefully analyze how their performance compares with finetuning using online RL algorithms. We show that finetuning with online RL may improve faster. Moreover, they typically load the offline data into the replay during finetuning. We show that doing so finetunes slowly.
> 2. Proposes additional changes, such as changing how data is sampled during finetuning [1]. However, when compared with offline RL algorithms for finetuning performance, they do not change the way the baselines samples data during finetuning. We show that even a simple online-only approach, allows offline RL baselines to finetune efficiently.

---

> ### Author Response · Authors · 2023-07-25
> **Comparison to RLPD**
>
> RLPD is a concurrent approach that demonstrates strong empirical performance for offline to online finetuning. We differ from RLPD in the following ways:
> 1. The default RLPD implmementation does not perform offline pretraining, thus policy collapse is not a concern in RLPD as the algorithm learns from scratch with the help of offline data.
> 2. RLPD uses online algorithm (REDQ) for finetuning and uses a fixed ratio of offline to online data for finetuning.  We compare two ways of performing online sampling and also included a setup similar to RLPD in the appendix. Overall we found that sampling more online data helps improve finetuning with standard offline RL algorithms.
>
> We included additional comparison with RLPD in the updated manuscript. We found that a large fraction of  performance improvement in RLPD comes from using a high UTD ratio (20). When using a comparable UTD (1), RLPD performs similarly or worse than what we presented in this paper.

---

### Decision · Action_Editors · 2023-08-19

**Recommendation:** Reject

**Comment:**

The paper studies the offline-to-online Reinforcement Learning setting.
The paper well motivates the importance of studying how to fine-tune policies learned from offline data.
The authors present a variety of experiments that show how the approach is reasonable and yield promising results.
However, the reviewers have pointed out several issues that, even after the authors' rebuttals, make this paper not strong enough to be published in TMLR.
In particular, the reviewers have found a lot of overlap with other published works that have not been referenced in this paper.
Even if the authors claim that their work was carried out before these other papers were published, now they cannot ignore them. To be published, this paper needs to position itself with respect to these recent papers too, and highlight what are the elements of novelty.
Beyond this, there are many issues related to the presentation of the results, the analysis of the algorithms, and a lack of discussion of their implications.

**Audience:**

The paper could be of interest to part of the TMLR-s audience.

**Claims And Evidence:**

The claims in the paper are not supported by enough empirical evidence.

**Resubmission Of Major Revision:**

The authors may consider submitting a major revision at a later time.